# *N*-Glycosylation in Piroplasmids: Diversity within Simplicity

**DOI:** 10.3390/pathogens10010050

**Published:** 2021-01-08

**Authors:** Monica Florin-Christensen, Anabel E. Rodriguez, Carlos E. Suárez, Massaro W. Ueti, Fernando O. Delgado, Ignacio Echaide, Leonhard Schnittger

**Affiliations:** 1Instituto de Patobiología Veterinaria (INTA-CONICET), CICVyA, Instituto Nacional de Tecnología Agropecuaria (INTA), Hurlingham 1686, Argentina; rodriguez.anabel@inta.gob.ar (A.E.R.); delgado.fernando@inta.gob.ar (F.O.D.); schnittger.leonhard@inta.gob.ar (L.S.); 2Consejo Nacional de Investigaciones Científicas y Técnicas (CONICET), Ciudad Autónoma de Buenos Aires C1033AAJ, Argentina; 3Department of Veterinary Microbiology and Pathology, College of Veterinary Medicine, Washington State University, Pullman, WA 99163, USA; suarez@wsu.edu (C.E.S.); massaro_ueti@wsu.edu (M.W.U.); 4Animal Disease Research Unit, United States Department of Agricultural-Agricultural Research Service, Pullman, WA 99163, USA; 5Estación Experimental Agrícola INTA-Rafaela, Santa Fe, Provincia de Buenos Aires S2300, Argentina; echaide.ignacio@inta.gob.ar

**Keywords:** piroplasmids, *Babesia*, *Theileria*, *Cytauxzoon*, *N*-glycan, dolichol, sugar nucleotides

## Abstract

*N*-glycosylation has remained mostly unexplored in Piroplasmida, an order of tick-transmitted pathogens of veterinary and medical relevance. Analysis of 11 piroplasmid genomes revealed three distinct scenarios regarding *N*-glycosylation: *Babesia* sensu stricto (s.s.) species add one or two *N*-acetylglucosamine (NAcGlc) molecules to proteins; *Theileria equi* and *Cytauxzoon felis* add (NAcGlc)_2_-mannose, while *B. microti* and *Theileria* s.s. synthesize dolichol-P-P-NAcGlc and dolichol-P-P-(NAcGlc)_2_ without subsequent transfer to proteins. All piroplasmids possess the gene complement needed for the synthesis of the *N*-glycosylation substrates, dolichol-P and sugar nucleotides. The oligosaccharyl transferase of *Babesia* species, *T. equi* and *C. felis*, is predicted to be composed of only two subunits, STT3 and Ost1. Occurrence of short *N*-glycans in *B. bovis* merozoites was experimentally demonstrated by fluorescence microscopy using a NAcGlc-specific lectin. In vitro growth of *B. bovis* was significantly impaired by tunicamycin, an inhibitor of *N*-glycosylation, indicating a relevant role for *N*-glycosylation in this pathogen. Finally, genes coding for *N*-glycosylation enzymes and substrate biosynthesis are transcribed in *B. bovis* blood and tick stages, suggesting that this pathway is biologically relevant throughout the parasite life cycle. Elucidation of the role/s exerted by *N*-glycans will increase our understanding of these successful parasites, for which improved control measures are needed.

## 1. Introduction

Apicomplexan protozoa are obligatory parasites, many of which seriously affect human and animal health. They undergo complex life cycles that frequently include host switches and morphological changes. Post-translational modifications (PTMs) of proteins are intricately involved in numerous aspects of their biology, including invasion of host cells, egress, differentiation, and protein localization, as well as modification of the host cell environment to create beneficial conditions for the parasites to complete their life cycle [1].

Among apicomplexan PTMs, there is scarce knowledge about *N*-glycosylation, one of the main eukaryotic modifications of proteins. *N*-glycosylation consists of the addition of an oligosaccharide of diverse composition and complexity to a surface-exposed Asn residue of a protein generally destined to the secretory pathway. Certain requisites are needed for this pathway, including: (i) A dolichol-P (dol-P) precursor on which the glycan is first assembled; (ii) an array of Alg (Asn-linked glycosylation) enzymes that take care of the sequential addition of monosaccharides to this dol-P precursor; (iii) the synthesis of nucleotide-activated sugars as substrates in the initial steps of *N*-glycosylation, and, but not essentially, lipid-bound sugars for the subsequent steps; (iv) a flippase in charge of transferring the lipid linked oligosaccharide (LLO) and in most cases, lipid-bound sugars as well, from the cytoplasmic to the luminal side of the rough endoplasmic reticulum (RER); (v) an oligosaccharide transferase that allocates the formed glycan on a nascent protein, and (vi) a surface-exposed Asn-X-Ser/Thr sequon in a nascent protein onto which the glycan is transferred, where X can be any residue but Pro. Independently of the final *N*-glycan structure, the first monosaccharides added to the sugar chain are always two *N*-acetylglucosamine (NAcGlc) residues, which are followed in most organisms by a two branch-structure of mannose (Man), glucose and sometimes other monosaccharide molecules. Only in higher eukaryotes, after *N*-glycan synthesis and transfer to a protein in the RER, remodeling of the *N*-glycan continues at the Golgi apparatus [2].

There is high diversity in *N*-glycan complexity among different organisms, which is determined by the repertoire of *N*-glycosylation enzymes encoded in their genomes [3]. Even among apicomplexans, *N*-glycans can vary from a basic *N*-acetyl chitobiose, i.e., (NAcGlc)_2_ in *Plasmodium falciparum*, to a more complex structure in which (NAcGlc)_2_ is linked to a bifurcated branch of 5 Man and 3 glucose residues in *Toxoplasma gondii* [3,4].

Diverse functions have been described for protein *N*-glycans in different organisms, some of which can be hypothesized to be relevant for apicomplexan parasites as well, such as protein folding, N-glycan-dependent ER-associated degradation of misfolded proteins (*N*-glycan ERAD), protection against protease cleavage, cell-cell interactions, and modulation of and protection from host immune responses [5]. Additionally, apicomplexan-specific functions, such as participation in gliding motility and in life stage differentiation have been proposed based on *N*-glycosylation inhibition experiments in *T. gondii* and *P. falciparum*, respectively [4,6].

*N*-glycosylation has remained poorly explored in piroplasmids, a vast group of tick-transmitted apicomplexan hemoprotozoans that include *Babesia*, *Theileria* and *Cytauxzoon* species, with relevance for veterinary and human health [7,8]. In this study, we analyze the repertoire of *N*-glycosylation-related genes in several piroplasmid species, unraveling interesting and previously undescribed differential features. We also present the first experimental evidence of the occurrence of a very basic form of *N*-glycan in *B. bovis*, one of the causative agents of bovine babesiosis, and explore its importance for parasite survival.

## 2. Results and Discussion

### 2.1. Three N-Glycosylation Scenarios Are Inferred from the Genomic Analysis of Diverse Piroplasmid Lineages

The repertoire of enzymes involved in *N*-glycosylation was identified in the predicted proteomes of 11 piroplasmid species for which genome sequences are available. The first step in this pathway consists of the assembly of a NAcGlc-P moiety, donated by the sugar nucleotide UDP-NAcGlc, on a membrane-inserted dol-P, to form dol-P-P-NAcGlc. This process takes place in the cytoplasmic side of the RER membrane, and is catalyzed by UDP-NAcGlc:dol-P *N*-acetylglucosamine phosphotransferase (Alg7, Figure 1). All of the analyzed piroplasmids were found to contain Alg7 in their proteomes, which implies that all are able to synthesize a lipid-linked NAcGlc at the RER membrane (Table 1).

The second step of *N*-glycosylation is the addition of another NAcGlc molecule to dol-P-P-NAcGlc, also using UDP-NAcGlc as donor. This process is catalyzed by a heterodimeric Alg13/Alg14 enzyme. Canonical versions of Alg13 and Alg14 could be predicted in nine and six of the analyzed piroplasmid genomes, respectively (Table 1).

Different *N*-glycosylation pathways result in three possible alternative scenarios exhibited by three different groups of piroplasmid lineages. *B. microti* and *Theileria* sensu stricto (s.s.) species (*T. annulata*, *T. parva*, *T. orientalis*) lack the genes coding for Alg1 and/or for oligosaccharide transferase (STT3). In other organisms, these two enzymes are responsible for adding additional Man residues and transferring the *N*-glycan onto a nascent protein, respectively. Thus, the end products in species belonging to *Theileria* s.s. are the LLO dol-P-P-(NAcGlc)_2_ and/or dol-P-P-NAcGlc. Importantly, transcriptomic data available for *T. parva* shows that Alg7 and Alg13 genes are transcribed in the schizont stage (data not shown), suggesting that these short LLO fulfill a functional role in these and other piroplasmids.

A gene encoding for an Alg1 enzyme is present exclusively in the *T. equi* and *C. felis* genomes (Table 1). This enzyme adds a Man moiety to the *N*-glycan, at the cytoplasmic side of the RER, using GDP-mannose as donor, to form dol-P-P-(NAcGlc)_2_Man. The shared ability of *C. felis* and *T. equi* to synthesize this compound correlates with their close phylogenetic relationship [8,9].

Most eukaryotic organisms add two branches with one and three Man residues to the pre-existing Man residue by means of the Alg2 and Alg11 glycosyl transferases, using GDP-Man as sugar donor [2]. However, piroplasmids lack this type of glycosyl transferases, thus, dol-P-P-NAcGlc, dol-P-P-(NAcGlc)_2_ and dol-P-P-(NAcGlc)_2_Man, the latter only in the case of *T. equi* and *C. felis*, are the only LLO produced at the cytoplasmic leaflet of the RER membrane. LLO need to be flipped across the RER membrane to the luminal side, a process shown to be protein-dependent and ATP-independent [10]. However, the flippase enzyme involved in this process remains unidentified for any organism at the molecular level.

In most eukaryotes, including several apicomplexans, glycosyl transferases further enlarge the *N*-glycan structure with Man and glucose moieties using dol-bound monosaccharides as donors that are present at the luminal side of the RER membrane [11]. Piroplasmids, as well as *Plasmodium* spp. and *Giardia lamblia*, lack this type of dol-P-sugar-dependent glycosyl transferases and consequently, any further modifications of their LLO (Table 1) [3].

In a final step, the oligosaccharide is transferred en bloc from the dolichyl pyrophosphate to the amide group of a selected Asn residue of a nascent protein (Figure 1). This process is catalyzed by a heterooligomeric complex of membrane proteins with oligosaccharyl transferase activity (OST). The OST complex is formed by eight subunits in yeast, six in *Cryptosporidium parvum*, four in *P. falciparum* and only one, the catalytic STT3 subunit, in *G. lamblia* and kinetoplastids [12]. In this study, we searched for the piroplasmid orthologs of the eight yeast OST subunits and found that *Babesia* s.s., *T. equi* and *C. felis* encode two of them, STT3 and Ost1, the latter also known as ribophorin 1 (Table 1). Notably, an OST complex formed exclusively by STT3 and Ost1 has not yet been described for any other organism so far. In contrast, *B. microti* and *Theileria* s.s. species, do neither encode STT3 nor Ost1 (Table 1). As exemplarily analyzed, the topology of STT3 of *B. bovis* follows that of other eukaryotic STT3, comprising a cytosolic N-terminal stretch followed by 13 transmembrane domains and, finally, a hydrophilic domain exposed to the RER lumen (Appendix A) [12].

Noteworthy, future studies of genome curation and gene re-annotation might help in the identification of genes that could not be found in this study, such as Alg13 and Alg14 of some piroplasmids. Indeed, the identification of *T. parva* Alg14, as well as indications for the presence of an Alg14 orthologue in the genome of *T. annulata*, were recently reported after re-examination of the genomes of these two *Theileria* spp. [13].

A great variety of LLO can act as donors for OST in different organisms. Importantly, it has been experimentally shown that dol-P-P-NAcGlc fulfills the minimum requirements to act as a glycosyl donor for OST, while the efficiency of transfer is significantly improved by addition of a second NAcGlc moiety, as in dol-P-P-(NAcGlc)_2_ [14].

Summarizing, our findings highlights three principal scenarios of *N*-glycan formation in piroplasmids: (i) *N*-glycosylation of proteins with (NAcGlc)_2_ or NAcGlc (*Babesia* s.s.); (ii) *N*-glycosylation of proteins with (NAcGlc)_2_Man (*T. equi* and *C. felis*); and (iii) synthesis of the LLO dol-P-P-NAcGlc and dol-P-P-(NAcGlc)_2_ that are not transferred to proteins (*B. microti* and *Theileria* s.s.). It can be observed that these three *N*-glycosylation enzyme sets can be assigned to the piroplasmid lineage of *Babesia* s.s. (Clade VI, set i), the closely related lineages of *T. equi* and *C. felis* (Clades IV and IIIa, respectively, set ii) and the distantly related *B. microti* and *Theileria* s.s. (Clades I and V, set iii) [8,9]. Similar to what was recently reported for piroplasmid C1A cysteine proteases and rhomboid proteinases, this observation corroborates the currently recognized phylogeny of piroplasmids [15,16]. On the other hand, the differences observed in the *N*-glycosylation pathway between *B. microti* and *Babesia* s.s., and between *T. equi* and *Theileria* s.s., underscore the need to revise the taxonomic placement of these two piroplasmids within the *Babesia* and the *Theileria* genera, respectively, as has been put forward before [8,9,17].

Notably, it has been shown that *Theileria* s.s. and *Babesia* s.s. merozoite blood stages invade host cells by entirely different mechanisms. The invasion of *Theileria* s.s. sporozoites is driven by the formation of a zipper between the parasite and the host cell membranes, and, subsequently, the parasite resides naked in the host cell cytoplasm [18]. In contrast, *Babesia* s.s. invade their host cells by formation of a tight moving junction, resulting in the formation of a parasitophorous vacuole which encases the parasite within the host cell cytoplasm, and then dissolves [19]. It remains to be investigated whether there is a relationship between the end products of the respective *N*-glycosylation pathways of these two piroplasmid lineages and the distinct mechanisms of host cell invasion.

The reduced repertoire of *N*-glycosylation enzymes found in piroplasmids as compared to higher eukaryotes could have resulted from secondary loss. This notion is in line with the hypothesis that all extant eukaryotes derive from a relatively complex common eukaryotic ancestor that lost some features along evolution due to adaptations to parasitism, among other events [20]. Moreover, it has been suggested that some kind of *N*-glycosylation was present in LUCA since both eubacteria and archaea also assemble oligosaccharides—though of different nature from their eukaryotic counterparts—onto a lipid precursor at their cellular membranes [21]. Consistently, we have observed that piroplasmid Alg7, Alg13, Alg14, and Alg1 are homologs of bacterial glycosyltransferases, similar to what has been reported for yeast Alg13 and Alg14 (results not shown) [22].

### 2.2. Piroplasmid Genomes Encode All Necessary Enzymes to Synthesize N-Glycan Substrates

Sugar nucleotides and dol-P are the only substrates used in the *N*-glycosylation pathway of piroplasmids. Regarding the first, UDP-NAcGlc is the donor for Alg7 and Alg13/Alg14, while GDP-Man is used as sugar donor for Alg1, which, as mentioned before, is present in *T. equi* and *C. felis*. The biosynthetic pathway of these nucleotide sugars was inferred in *P. falciparum* from the enzyme-coding genes identified in its genome and experiments using radiolabeled sugars [23]. The precursor of UDP-NAcGlc synthesis is glucose-6P, that undergoes conventional eukaryotic de novo routes, in which one of the intermediaries is glucosamine-6P (Glc-6P). Additionally, a salvage pathway involves the incorporation of glucose, fructose or Glc from the medium and their conversion to Glc-6P by hexokinase (HK). Glc-6P undergoes *N*-acetylation, forming NAcGlc-6P which is in turn transformed to GlcNAc-1P. Finally, UDP-*N*-acetylglucosamine pyrophosphorylase takes care of the formation of the nucleotide sugar with involvement of UTP. GDP-Man synthesis, on the other hand, starts with the conversion of fructose-6-P into Man-6-P, or can also follow a salvage route, with the uptake of Man from the medium and its transformation to Man-6P by HK. Man-6P is then converted to Man-1-P which, by association with a GTP molecule, catalyzed by Man-1-phosphate guanyltransferase, yields GDP-Man and pyrophosphate (Appendix A).

Noteworthy, reverse BLAST showed that orthologues of *P. falciparum* genes implicated in the biosynthetic pathways of UDP-NAcGlc and GDP-Man are present in all piroplasmid genomes studied in this work. The corresponding enzymes of *B. bovis*, *T. equi* and *T. parva*, which represent the three different *N*-glycosylation scenarios revealed in this study, are listed in Appendix A. It is important to note that all piroplasmids use GDP-Man as sugar donor for dol-P sugars in the biosynthesis of glycosylphosphatidylinositol (GPI), while exclusively *T. equi* and *C. felis* use this sugar nucleotide also for *N*-glycosylation [24].

Dol, a cis-polyisoprenoid, in which the terminal isoprene unit is saturated and oxidized to an alcohol, is a substrate in both the *N*-glycosylation and the GPI biosynthetic pathways. In apicomplexan parasites, as well as in eubacteria and plant chloroplasts, isoprenoid precursors are synthesized through the methylerythritol phosphate (MEP) pathway, while in other eukaryotes, synthesis takes place through the mevalonate pathway. The MEP pathway is a seven-step process that starts with pyruvate and ends with the synthesis of isopentenyl diphosphate (IPP) and dimethylallyl diphosphate (DMAPP), and, in apicomplexans, takes place in the apicoplast [25]. All MEP pathway enzymes were shown to be encoded in the genomes of *B. bovis*, *B. microti*, and *B. orientalis*, and expression of two of them (BoDXS and BoDXR) was experimentally demonstrated in merozoites (mz) of the latter [26,27]. The MEP pathway is absent in mammals, and inhibitors of this route hamper the in vitro growth of *Babesia* spp. mz, making it an attractive therapeutic target against piroplasmids [26,27,28].

It has been proposed for *P. falciparum* that the end products of the MEP pathway, IPP and DMAPP, are converted outside the apicoplast into farnesyl diphosphate (FPP) and geranylgeranyl diphosphate (GGPP), and that the final steps of dol-P synthesis take place at the cytoplasmic leaflet of the RER membrane [25]. Thus, some kind of transport system needs to carry these hydrophobic molecules through the hydrophilic cytoplasmic milieu. There, an isoprenoid elongation step, catalyzed by cis-prenyltransferase (CPT) takes place, resulting in a polyprenyl diphosphate bearing a varying number of isoprene units which, in *P. falciparum*, has been determined to be 15 to 19. Dephosphorylation by polyprenyl diphosphate phosphatase yields polyprenol, which is then reduced by polyprenol reductase (PPRD) to dol [25]. Dol needs then to undergo phosphorylation to be utilized in the *N*-glycosylation and GPI synthesis pathways. In yeast, this step is carried out by SEC59 dolichol kinase (NP_013726.1). Homology searches using *P. falciparum* CPT and PPRD as queries showed that all piroplasmid genomes analyzed encode these enzymes. Also, SEC59 homologues were found in all piroplasmids (Appendix A).

In addition to de novo synthesis, evidences of dol recycling have been obtained in yeast. After transfer of the glycan to an Asn residue of a nascent protein, dol-P-P was shown to be dephosphorylated yielding dol-P, which was flipped to the cytoplasmic leaflet of the RER membrane, becoming thus available for reuse [29]. This dol-P flipping step likely also takes place in piroplasmids (Figure 1). In addition to its use in *N*-glycosylation and GPI biosynthesis, dol was demonstrated to be covalently added to terminal cysteine residues of *P. falciparum* proteins [30]. It remains to be determined whether piroplasmids also possess this type of PTM.

In short, the genome searches carried out in this study show that all analyzed piroplasmids possess the necessary machinery to synthesize *N*-glycan substrates independently from their hosts.

### 2.3. The N-Glycan N-Acetyl-chitobiose Can Be Detected on the B. bovis mz Surface

To experimentally examine if the predicted *N*-glycosylation pathway is operational in the bovine piroplasmid *B. bovis*, smears of infected erythrocytes were incubated with biotin-conjugated *Griffonia simplicifolia* Lectin II (GSL II), followed by detection with Fluorescein Isothiocyanate (FITC)-labeled streptavidin and observation in a fluorescent microscope. GSLII is a dimeric glycoprotein that specifically binds α- or β-linked NAcGlc residues on the non-reducing terminal end of oligosaccharides. The labeling of *B. bovis* mz observed with this lectin is consistent with the in silico prediction of the occurrence of short *N*-glycans, composed of one or two NAcGlc moieties, modifying the surface proteins of this parasite (Figure 2). Additionally, LLO associated to parasite membranes might become labeled as well. The fluorescent pattern obtained shows diffuse labeling that corresponds to the parasite surface and some strong spots consistent with RER membrane labeling.

### 2.4. Abundant Proteins of the B. bovis Secretome Are Predicted to Be N-Glycosylated

The predicted secretome of *B. bovis* that comprises all signal peptide-bearing proteins, was searched to identify proteins that potentially undergo *N*-glycosylation, i.e., that carry one or more surface-exposed Asn-X-Ser/Thr sequons, where X can be any amino acid but Pro. Only sites with a high prediction for the presence of a signal peptide and *N*-glycosylation were considered, resulting in the identification of 13 membrane-bound and 50 exported proteins (Appendix A). Apart from a large number of “hypothetical” proteins, the list includes proteins shown to participate in host-pathogen interactions, such as 12D3 antigen, thrombospondin-related anonymous protein (TRAP) and spherical body protein 3 (SBP3), the first two of which also constitute vaccine candidates [31,32,33]. Notably, SBP3 has been experimentally shown to belong to the exportome, and to be localized in the erythrocyte cytosol [34]. Additionally, several smORFs that belong to a large family of variable proteins of still unknown function are also likely to be *N*-glycosylated [35]. Most of the identified proteins carried a single highly predicted *N*-glycosylation site, whereas the presence of 2 or up to 7 sites was less frequent (Appendix A).

Importantly, however, in silico prediction of *N*-glycosylation does not necessarily correspond to the situation encountered in vivo. It has been shown for other organisms that up to 35% of proteins bearing an *N*-glycosylation site might not be glycosylated at the RER [36,37]. Actual addition of an *N*-glycan depends on the position of the sequon, being less frequent on those close to a transmembrane domain, the C-terminus or a disulfide bridge [12]. In *Trypanosoma brucei*, it was shown that the presence of certain amino acids in the proximity of a sequon critically influences the *N*-glycosylated status of a protein [38]. Additionally, *N*-glycosylation of a particular protein can vary with the life cycle stage, as shown for *P. falciparum* [6].

In conclusion, a considerable number of *B. bovis* proteins bear the necessary features to be N-glycosylated, but the actual presence of *N*-glycans in specific proteins and at particular life stages remains to be experimentally demonstrated. Also, the presence of *N*-glycosylation sequons has to be taken into account when producing recombinant proteins in eukaryotic systems, since abnormal *N*-glycosylation patterns are likely to occur [36].

### 2.5. N-Glycosylation Enzyme Genes Are Transcribed in B. bovis Blood and Tick-Kinete Parasite Stages

We compared the levels of transcription of *B. bovis* genes involved in *N*-glycosylation -related pathways between bovine blood (mz) and tick hemolymph (kinetes) parasite stages. The normalized data are shown in Figure 3 and Appendix A. None of the genes coding for enzymes involved in the *N*-glycosylation pathway have a differential pattern of expression among blood and kinete stages, with the exception of transcripts for the Alg13 gene which were significantly higher in blood stages. Some of the genes showed marginal, but still detectable, levels of expression in both (Figure 3A). Similarly, no significant differences in the levels of transcripts were found for the two genes involved in the synthesis of dol (Figure 3B).

In contrast, differential transcription levels of genes involved in UDP-NAcGlc synthesis in *B. bovis* blood and tick-stage parasites seems to indicate the favoring of a different pathway in each of these stages. While blood stages have a marked increased transcription of the genes involved in de novo synthesis of UDP-NAcGlc from glucose-6-P, G6PI, and GFPT, transcription of the gene coding for HK, the hexokinase that participates in the scavenging of Glc from the milieu by transforming it to Glc-6-P, is substantially increased in the tick stages. Expression of PAGM, that transforms NAcGlc-6P into NAcGlc-1P before its coupling to the UDP nucleotide is also significantly increased at the transcriptional level in tick stages (Figure 3B and Appendix A).

As mentioned before, UDP-NAcGlc is the only sugar nucleotide substrate used for *N*-glycan synthesis in *B. bovis*, while GDP-Man acts as sugar donor exclusively in GPI biosynthesis. Thus, we explored whether the transcription level differences between blood and tick stages observed in the UDP-NAcGlc biosynthetic pathway were also observed in the case of the GDP-Man biosynthetic pathway. Surprisingly, transcription of the genes encoding MPI, PMM, and MPG that participate in GDP-Man synthesis are all increased (8, 70, and 2-fold, respectively) in tick—as compared to blood stages, following the same pattern of HK, which also participates in the scavenging of Man that allows its incorporation to this route (Figure 3C and Appendix A).

Our results indicate that *N*-glycosylation of proteins as well as synthesis of *N*-glycosylation substrates is operative in both blood and tick stages. They also highlight that synthesis of the sugar nucleotide precursor for GPI synthesis, GDP-Man, could be of particular importance for the tick stage. The biological significance of the observed differences in transcription levels is worth exploring in future research.

### 2.6. N-Glycosylation Is Important but Not Essential for In Vitro Growth of B. bovis mz

As a first step towards evaluating the biological relevance of piroplasmid *N*-glycosylation, the effects of tunicamycin (Tun) on the in vitro growth of *B. bovis* merozoites was analyzed. Tun inhibits *N*-glycosylation in a wide variety of organisms, from archaea to higher eukaryotes, by blocking the transfer of UDP-GlcNAc to dol-P by specific interaction with Alg7 [39,40,41]. *B. bovis* cultures exposed to growing concentrations of Tun (0 to 30 μM) for 72 h showed a dose-dependent decrease in percentages of parasitized erythrocytes (PPE) (Figure 4). These results agree well with those observed for *P. falciparum* and *T. gondii* treated with similar concentrations of this drug [4,6,42]. Despite a significant impairment of *B. bovis* growth in the presence of Tun, a considerable number of parasites were still able to reproduce, suggesting an important but not essential role for *N*-glycosylation in erythrocyte invasion and parasite survival, at least in in vitro cultures. It remains to be determined whether Tun-treated parasites are able to develop in vivo and produce acute and persistent infection, develop sexual stages and reproduce sexually in the tick vector, and whether these parasites have an attenuated phenotype.

The role/s of *N*-glycosylation in piroplasmids and other apicomplexans remains to be elucidated. In higher eukaryotes, *N*-glycans participate in quality control of protein folding (*N*-glycan QC) and *N*-glycan ERAD. However, the short glycans present in piroplasmids cannot fulfill these roles, indicating that *N*-glycan QC and ERAD are not needed for piroplasmid viability [43].

In *T. gondii*, Tun-treated parasites showed defective motility, suggesting that *N*-glycosylation could be involved in glideosome function, an essential feature to ensure host cell invasion and survival [4,44]. In *P. falciparum*, on the other hand, Tun impaired the differentiation of trophozoites to the schizont stage by still undefined mechanisms [6].

Our observations that transcription of *N*-glycosylation genes occurs in *B. bovis* parasite life stages that live either inside bovine erythrocytes or in the tick hemolymph suggest that this process could be related to motility, invasion, differentiation, and/or other mechanisms needed for parasitic life in these environments. However, this speculation needs experimental confirmation. Rapidly developing genetic tools could be used in the future to carry out loss of function experiments in order to locate the process or processes for which *N*-glycosylation is critical for piroplasmids [45].

## 3. Materials and Methods

### 3.1. Bioinformatic Analysis

Homologous genes to those encoding the enzymes that participate in the *N*-glycosylation pathway, as well as in sugar nucleotide synthesis and the final steps of dol-P synthesis of *Saccharomyces cerevisiae* S288C and/or *P. falciparum* strain 3D7 [3,23,25,46] were searched for in the available genomes of *B. bovis* strain T2Bo [47], *B. bigemina* strain BOND [48], *B. divergens* strain 1802A [49], *B. ovata* strain Miyake [50], *Babesia* sp. Xinjang [51], *B. microti* strain R1 [52], *T. parva* strain Muguga [53], *T. annulata* strain Ankara [54], *T. orientalis* strain Shintoku [55], *T. equi* strain WA [17], and *C. felis* strain Winnie [56]. The algorithms of NCBI BLAST (blast.ncbi.nlm.nih.gov/Blast.cgi), KEGG Pathway Database (www.genome.jp/kegg/pathway.html), and PiroplasmaDB (piroplasmadb.org/piro/) were used. Conserved domains and transmembrane domains of individual proteins were verified in the NCBI Conserved domains database (www.ncbi.nlm.nih.gov/Structure/cdd/wrpsb.cgi?) and TOPCONS (http://topcons.net/pred/), respectively. For the analysis of the whole predicted proteome of *B. bovis*, T2Bo strain, TOPCONS (topcons.net/pred/) and DeepLoc (www.cbs.dtu.dk/services/DeepLoc/) were used to differentiate exported, soluble and membrane-bound proteins. NetNGlyc 1.0 Server (www.cbs.dtu.dk/services/NetNGlyc/) was used to predict *N*-glycosylation sites with a probability of 0.75 or higher. In the case of transmembrane proteins, only those with surface-exposed *N*-glycosylation sites were chosen.

### 3.2. Parasite Cultures

*B. bovis* mz of the Argentine S2P pathogenic strain [57] were in vitro cultured on bovine erythrocytes, in 75 cm^2^ tissue culture bottles, essentially as described by Levy and Ristic [58]. The basic culture medium (BCM) consisted in M199 (Sigma-Aldrich, St. Louis, MO, USA)/5 mM HEPES (Sigma-Aldrich), 100 μg/mL streptomycin, 100 IU/mL penicillin, pH 7.2. This medium was supplemented with normal bovine serum from the same donor of erythrocytes in a BCM/serum proportion of 60:40 (*v/v*) (BCM-S). Cultures were grown on 10% packed bovine erythrocytes/90% BCM-S (*v/v*), at 37 °C, in a 5% CO_2_ atmosphere, with daily changes of BCM-S, using a starting PPE of 1%. Growth was evaluated by microscopic observation of Giemsa-stained smears (1000×). For maintenance, subcultures were started when PPE reached 3%.

### 3.3. Detection of N-Glycosylation by Fluorescence Microscopy

Smears of *B. bovis* S2P-infected erythrocytes were prepared, dried at room temperature for 30 min, washed once in phosphate-buffered saline (PBS) and fixed with cold acetone for 1 min. After washing with PBS, smears were blocked with 1% (*w*/*v*) bovine serum albumin in PBS for 30 min, followed by overnight incubation with 2 μg/mL biotinylated GSL-II (Vector Laboratories, Burlingame, CA, USA) or PBS as a negative control, at 4 °C. After two washes with PBS/0.05% Tween-20 for 10 min, slides were incubated with FITC-labeled streptavidin conjugate (Invitrogen, Waltham, MA, USA), diluted 1:1000 for 20 min at room temperature in the dark. *B. bovis* nuclei were labeled by incubation for 5 min with 2 μg/mL DAPI, followed by several washes with PBS. Slides were examined by fluorescence microscopy in a Nikon Eclipse 80i microscope at 1000× magnification.

### 3.4. Transcriptomics Analysis

The analysis of gene regulation during *B. bovis* infection of mammalian host and tick vector was performed by using RNA-Seq data sets deposited in the NCBI Gene Expression Omnibus (GEO; https://www.ncbi.nlm.nih.gov/geo/query/acc.cgi?acc=GSE144066) under accession number GSE144066. Briefly, the datasets were generated from *B. bovis* blood stages isolated from infected calves and *B. bovis* kinete stages from female *Ripicephalus microplus* tick hemolymph as previously described [59]. Evaluation of up-regulation of the genes of interest was compared by the log_10_ CPM of transcripts of the two *B. bovis* stages from vertebrate and invertebrate hosts.

### 3.5. Inhibition Tests

A *B. bovis*-infected erythrocyte suspension in BCM-S was distributed in 1 mL aliquots in the 6 central wells of 24-well plates with an initial PPE of approximately 1.5%, while the external wells were filled with 1 mL sterile distilled water. At T0, all suspensions received 20 μL of M199 medium containing different amounts of Tun prepared from a 1 mM stock in M199, so that the final Tun concentrations were 0, 0.3, 1, 3, and 10 μM. Each concentration was tested in triplicate wells. A 700 μL aliquot of supernatant was daily removed from each well and replaced with an equal amount of BCM-S containing the corresponding amount of Tun. After 72 h, PPEs were determined in Giemsa-stained smears, counting a minimum of 3000 erythrocytes/slide. The statistical significance of the PPE differences between treated and untreated cultures was estimated using the Student’s *t*-test.

## 4. Conclusions

In summary, our work provides new information about the occurrence and biological relevance of *N*-glycosylation in *B. bovis* and predicts the types of *N*-glycans and/or LLO present in this and other piroplasmids. Knowledge on the type, extent and biological relevance of *N*-glycosylation in these parasitic protozoa can help in the understanding of their survival strategies and pathogenicity mechanisms and in the design of control tools to alleviate the significant burden they impose on human and animal health.

## Figures and Tables

**Figure 1 pathogens-10-00050-f001:**
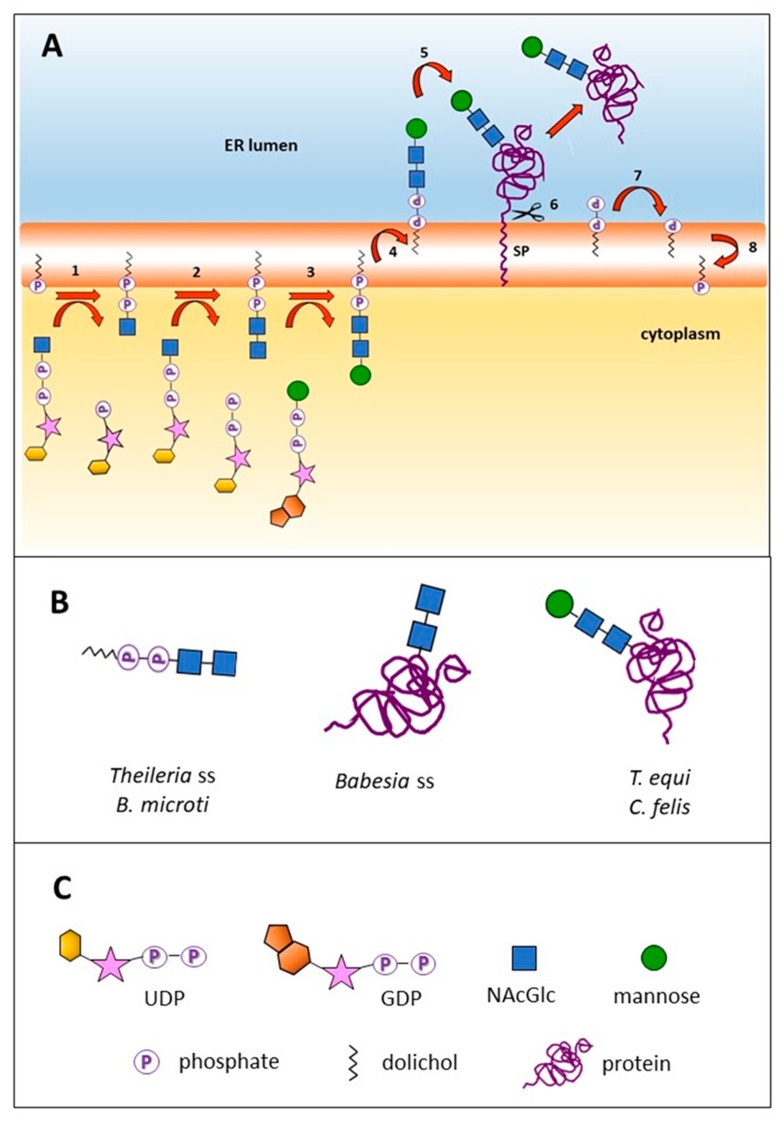
*N*-glycosylation pathways and end products in piroplasmids. (**A**) Predicted pathway for the *N*-glycosylation of an exported protein in *Theileria equi* and *Cytauxzoon felis*: (1) Synthesis of dol-P-NAcGlc by Alg7; (2) synthesis of dol-P-P-(NAcGlc)_2_ by Alg13/Alg14 (both processes occur at the cytoplasmic leaflet of the rough endoplasmic reticulum (RER) membrane and utilize UDP-NAcGlc as sugar donor); (3) synthesis of dol-P-P-(NAcGlc)_2_-Man by Alg1; (4) flipping to the luminal leaflet of the RER by a flippase; (5) transfer of (NAcGlc)_2_ to a signal peptide-bearing nascent protein by oligosaccharyl transferase activity (OST); (6) cleavage of signal peptide liberating the recently synthesized *N*-glycosylated protein (green); (7) dephosphorylation of dol-P-P to dol-P; (8) flipping of dol-P to the cytoplasmic leaflet of the RER for reuse. *Babesia* s.s. lack step 3, while *Theileria* s.s. and *B. microti* lack steps 3 and 5. SP: Signal peptide. (**B**) End products resulting from three different *N*-glycosylation enzyme sets: dol-P-P-(NAcGlc)_2_, and (NAcGlc)_2_ or Man-(NAcGlc)_2_ modifying a protein. (**C**) graphical references.

**Figure 2 pathogens-10-00050-f002:**
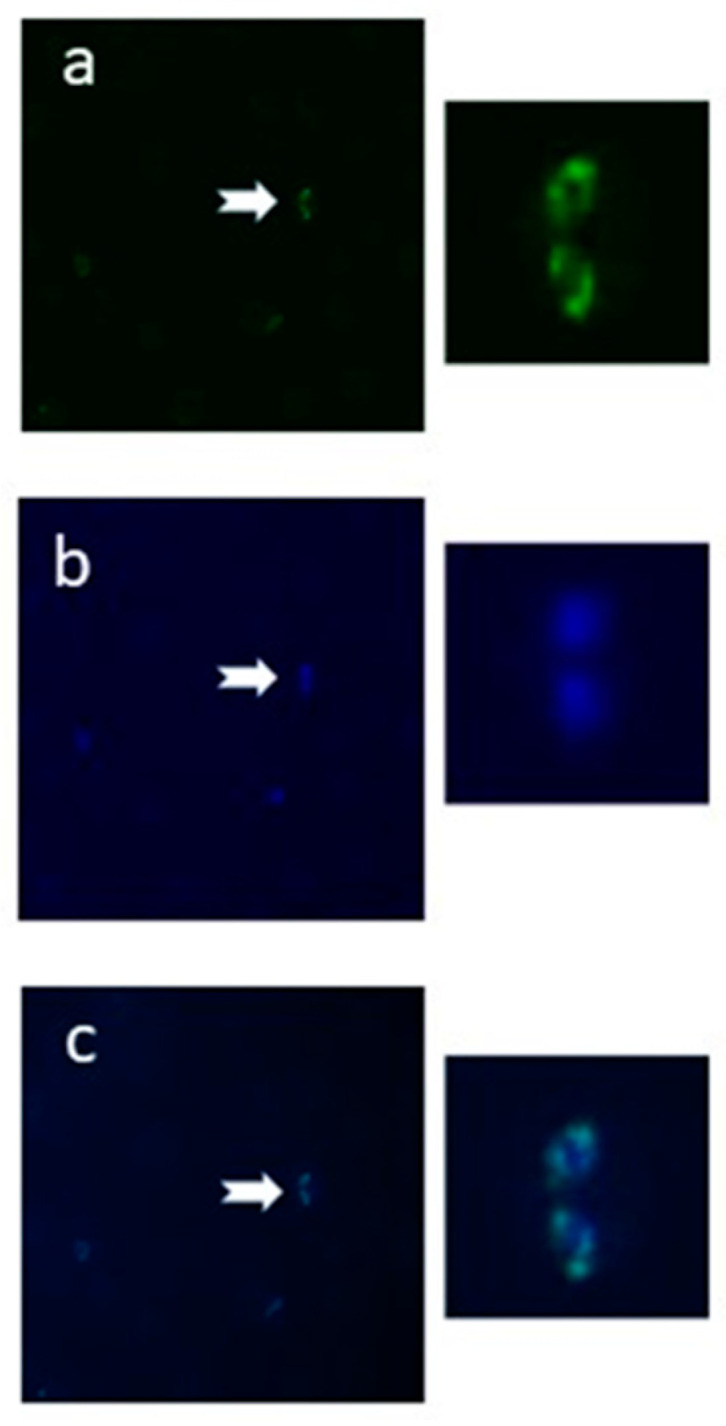
Detection of short NAcGlc glycans in *B. bovis* mz. Fixed smears of *B. bovis*-infected erythrocytes were incubated with biotin-labeled GSLII lectin, followed by FITC-streptavidin and 4,6-diamidino-2-phenylindole (DAPI) for nuclei Scheme 1000× in an epifluorescence microscope using filters to detect FITC (**a**) or DAPI (**b**). (**c**) Merged image.

**Figure 3 pathogens-10-00050-f003:**
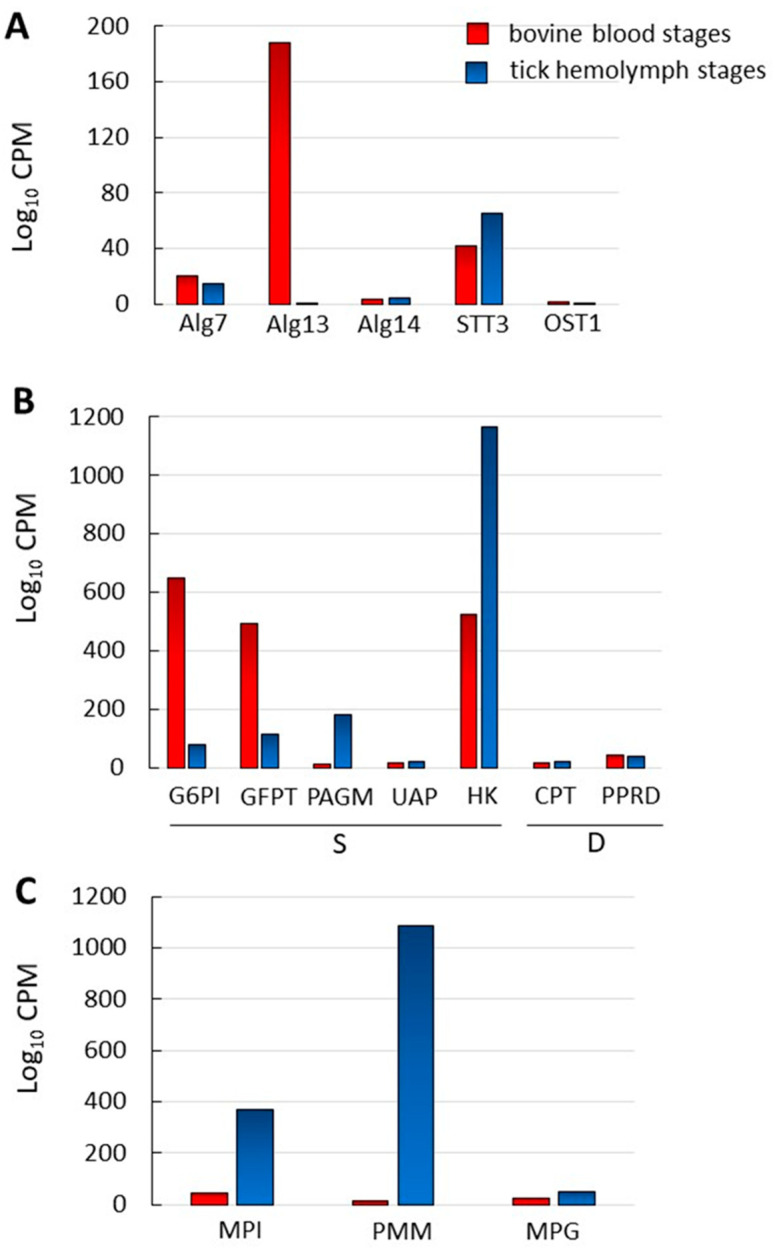
Comparison of the transcription levels of *N*-glycosylation -related genes in blood and tick *B. bovis* stages. (**A**) *N*-glycosylation pathway-encoding genes. (**B**) Genes related to the biosynthesis of UDP-NAcGlc (S) and dol (D), the substrates for *N*-glycan synthesis in *B. bovis*. (**C**) Genes related to the synthesis of GDP-Man, which in *B. bovis* is only used as sugar donor for GPI biosynthesis. Red bars: Blood stages; blue bars: Tick stages.

**Figure 4 pathogens-10-00050-f004:**
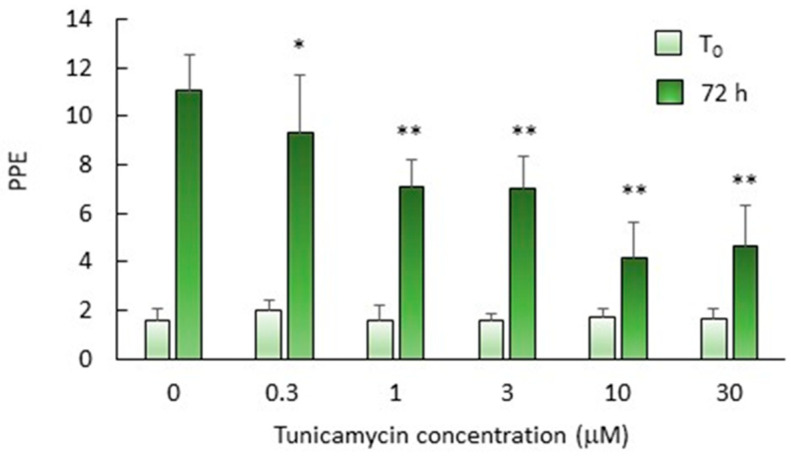
Dose-dependent effect of Tun on the in vitro growth of *B. bovis* merozoites. Percentages of parasitized erythrocytes (PPE) were calculated by microscopic observation of Giemsa-stained smears withdrawn at T0 (light blue bars) and after 72 h (dark green bars) of cultivation. Bars show the average + SD of three independent cultures. Asterisks mark significant decreases with respect to the untreated cultures (0 μM): * *p* < 0.05; ** *p* < 0.001.

**Table 1 pathogens-10-00050-t001:** Enzymes involved in the *N*-glycosylation pathway of different piroplasmids.

Set	Lineage	Activity	Addition of:				*N*-Glycan Transfer	
			1st NAcGlc	2nd NAcGlc		Man		
		Enzyme Species	Alg7	Alg13	Alg14	Alg1	STT3	OST1
i	Clade IV	*T. equi*	XP_004830167	XP_004829088	XP_004830080	XP_004832122	XP_004832354	XP_004831427
	Clade IIIa	*C. felis*	CF000673	CF001791	CF002639	CF004094	CF001242	CF000008
ii	Clade VI *Babesia* s.s.	*B. bovis*	XP_001609261	XP_001610083	XP_001610338	-	XP_001609551	XP_001611482
		*B. bigemina*	XP_012769992	XP_012766341	-	-	XP_012768251	XP_012769449
		*B. divergens*	Bdiv_013430c	Bdiv_003540c	Bdiv_029110	-	Bdiv_030100	Bdiv_036510
		*Babesia* sp. Xinjang	XP_028870826	XP_028872191	-	-	XP_028869940	-
		*B. ovata*	-	BOVATA_021750		-	BOVATA_014210	BOVATA_008000
iii	Clade I	*B. microti*	XP_021337988	-	-	-	-	-
	Clade V *Theileria* s.s.	*T. annulata*	XP_954592	-	-	-	-	-
		*T. parva*	XP_765645	XP_765081	KAF5153350.1	-	-	-
		*T. orientalis*	XP_009688943	XP_009690544	XP_009689017	-	-	-

## Data Availability

The data presented in Figure 3 are available in Appendix A. All other data are available on request from the corresponding author.

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
