# Peer review of "N-Glycosylation in Piroplasmids: Diversity within Simplicity"

_pathogens, 2021, doi:10.3390/pathogens10010050_

Round 1

Reviewer 1 Report

This study describes the biological relevance of N-glycosylation in B. bovis and predicts the type of N-glycans and/or LLO present in this and other piroplasmids.

Overall, this study provides evidence in the understanding of the survival strategies and pathogenicity mechanisms of these piroplasmids. What is more, it also might suggest how to specifically design tools aimed at reducing the several hazards inflicted to human and animal beings.

Author Response

This study describes the biological relevance of N-glycosylation in B. bovis and predicts the type of N-glycans and/or LLO present in this and other piroplasmids.

We are very grateful for the reviewer's revision and comments. A thorough revision of the text has been carried out. 

Reviewer 2 Report

The work is very interesting and provides novel information to the reader of this specific field. I would recommend to provide some further practical information on the use of your results and how they can improve the control of piroplasms. Some parts would be better be more explained in order to be understood better.

Author Response

We thank very much the reviewer for his/her revision of the manuscript and comments. We have made a thorough revision of the text for this second version. 

Reviewer 3 Report

This study was well designed and presented. However, I just have some minor comments.

Abbreviation: When you use an abbreviation, you have to use the abbreviated form from the second appearance, e.g., RER in L99, but it was first appeared in L58; PPE in L470, but first appeared in L432; FITC in L481, but appeared in L300; etc. Please give a full name with an abbreviation at the first appearance and then use only the abbreviated form.

Please use abbreviation when it is repeated, e.g., mz in L461, RT in L475, BSA in L477. If there is no repeat, please do not abbreviate. Please check this throughout the manu.

Abbreviation is also applied to the scientific names. When scientific names were repeated, the genus names should be abbreviated. Please refer to L439-459. Multiple species names were written with full names while they were repeated multiple times.

Vendor info: For a vendor in the USA: Description (Product name; Company name, City, State abbreviation, Country), or Product name (Company name, City, State abbreviation, Country). For a vendor outside the USA: Description (Product name; Company name, City/Town, Country), or Product name (Company name, City/Town, Country). When a vendor was repeated, please give only company name. Please refer to L439-509 for correct vendor description.

Minor comments:

L4: Please indicate the meaning of “&” in the affiliation.

L120: “anAlg1” to “an Alg1” for a spacing.

L121: “genomes. (Table 1).” to “genomes (Table 1).” for usage of period.

L140: Giardia lamblia for italics.

L183: “our findings highlights” to “our findings highlight” for the singular form of a verb.

L265: [26,27,28] to [26-28]

L326: [31,32,33] to [31-33]

L369: (Fig. 3B, Fig. S2) to (Figs. 3B, S2)

L379: (Fig. 3C, Fig. S2) to (Figs. 3C, S2)

L398: [39, 40,41] to [39-41]

L410-411: Please check this sentence for correctness.

Overall, description is okay, but the revised version should be edited by a native speaker to make sure the English is of high quality.

Author Response

1) Reviewer: This study was well designed and presented.

Answer: We thank the reviewer for his/her thorough revision of our manuscript

2) Reviewer: However, I just have some minor comments.

Abbreviation: When you use an abbreviation, you have to use the abbreviated form from the second appearance, e.g., RER in L99, but it was first appeared in L58; PPE in L470, but first appeared in L432; FITC in L481, but appeared in L300; etc. Please give a full name with an abbreviation at the first appearance and then use only the abbreviated form.

Answer: The ms has been checked for all abbreviations and those mentioned by the reviewer as well as others such as dolichol (dol) or tunicamycin (Tun) have been correctly incorporated.

3) Reviewer: Please use abbreviation when it is repeated, e.g., mz in L461, RT in L475, BSA in L477. If there is no repeat, please do not abbreviate. Please check this throughout the manu.

Answer: Merozoites has been abbreviated to mz, while abbreviations for Room temperature and bovine serum albumin have been deleted.

4) Reviewer: Abbreviation is also applied to the scientific names. When scientific names were repeated, the genus names should be abbreviated. Please refer to L439-459. Multiple species names were written with full names while they were repeated multiple times.

Answer: We have checked all species names so that they appear correctly abbreviated. The only exception is Babesia sp. Xinjang, which is fully spelled in the bibliography. 

5) Reviewer: Vendor info: For a vendor in the USA: Description (Product name; Company name, City, State abbreviation, Country), or Product name (Company name, City, State abbreviation, Country).

For a vendor outside the USA: Description (Product name; Company name, City/Town, Country), or Product name (Company name, City/Town, Country). When a vendor was repeated, please give only company name. Please refer to L439-509 for correct vendor description. 

Answer: This has been done

6) Reviewer: Minor comments:

L4: Please indicate the meaning of “&” in the affiliation. 

L120: “anAlg1” to “an Alg1” for a spacing. 

L121: “genomes. (Table 1).” to “genomes (Table 1).” for usage of period. done

L140: Giardia lamblia for italics. 

L183: “our findings highlights” to “our findings highlight” for the singular form of a verb. 

L265: [26,27,28] to [26-28] 

L326: [31,32,33] to [31-33] 

L369: (Fig. 3B, Fig. S2) to (Figs. 3B, S2) 

L379: (Fig. 3C, Fig. S2) to (Figs. 3C, S2) 

L398: [39, 40,41] to [39-41]

Answer: All these corrections have been incorporated.

7) Reviewer: L410-411: Please check this sentence for correctness.

Answer: The sentence has been revised and corrected.

8) Reviewer: Overall, description is okay, but the revised version should be edited by a native speaker to make sure the English is of high quality.

Answer: The manuscript was thoroughly revised and some corrections to improve the English were included. Revision by a native speaker was not possible given the timing assigned by the journal. However, we believe that the ms has been significantly improved by the corrections pointed out by the reviewer and a few additional small modifications that were introduced.